# Small Molecule Compounds, A Novel Strategy against *Streptococcus mutans*

**DOI:** 10.3390/pathogens10121540

**Published:** 2021-11-25

**Authors:** Sirui Yang, Jin Zhang, Ran Yang, Xin Xu

**Affiliations:** 1State Key Laboratory of Oral Diseases, National Clinical Research Center for Oral Diseases, Chengdu 610041, China; 2019224035135@stu.scu.edu.cn (S.Y.); zhangjin0831@stu.scu.edu.cn (J.Z.); 2Department of Cariology and Endodontics, West China Hospital of Stomatology, Sichuan University, Chengdu 610041, China; 3Department of Pediatric Dentistry, West China Hospital of Stomatology, Sichuan University, Chengdu 610041, China

**Keywords:** small molecules, *Streptococcus mutans*, drug repurposing, sortase A, glucosyltransferases (Gtfs)

## Abstract

Dental caries, as a common oral infectious disease, is a worldwide public health issue. Oral biofilms are the main cause of dental caries. *Streptococcus mutans* (*S. mutans*) is well recognized as the major causative factor of dental caries within oral biofilms. In addition to mechanical removal such as tooth brushing and flossing, the topical application of antimicrobial agents is necessarily adjuvant to the control of caries particularly for high-risk populations. The mainstay antimicrobial agents for caries such as chlorhexidine have limitations including taste confusions, mucosal soreness, tooth discoloration, and disruption of an oral microbial equilibrium. Antimicrobial small molecules are promising in the control of *S. mutans* due to good antimicrobial activity, good selectivity, and low toxicity. In this paper, we discussed the application of antimicrobial small molecules to the control of *S. mutans*, with a particular focus on the identification and development of active compounds and their modes of action against the growth and virulence of *S. mutans*.

## 1. Introduction

Dental caries is a chronic infectious disease across all ages of human beings [1], which seriously endangers human oral and general health and affects the quality of life [2]. Under normal conditions, the oral flora maintains a symbiotic relationship with the host [3]. However, under cariogenic conditions, such as frequent sugar intake, cariogenic bacteria compete with oral commensals and cause microbial dysbiosis. The dysbiosis of oral biofilm metabolizes carbohydrates and produces excessive acid, leading to pH declination and consequently tooth demineralization and tooth decay [4,5,6]. Among oral biofilms, *Streptococcus mutans* (*S. mutans*) is well recognized as the major cariogenic species due to its acidogenicity and aciduricity. Besides, *S. mutans* synthesizes exopolysaccharides (EPSs), which mediate the adhesion between cells and the tooth surface and contribute to the formation of oral biofilms and the development of dental caries [7,8]. Compared to planktonic cells, microbial biofilms show higher tolerance to acid and higher resistance to antimicrobial drugs [9]. Therefore, the control of *S. mutans*, particularly in its biofilm forms, is in great urgency.

Mechanical plaque removal and the application of chemotherapeutics are commonly used for the control of dental caries. Daily mechanical plaque control including tooth brushing and flossing is commonly used at all age groups for the prevention of dental caries. However, in the high-risk group for caries, the topical application of antimicrobials is necessary [10]. Broad-spectrum antimicrobials such as chlorhexidine digluconate (CHX) are widely used to control cariogenic pathogens [11]. However, CHX has limitations such as taste confusions, mucosal soreness, tooth discoloration, and drug resistance [12,13]. Therefore, new strategies or agents to control caries are needed. Small molecules are compounds with a molecular weight of less than 1000 Da [14]. Recently, small molecules have become promising alternatives for the control of oral biofilms due to good cell permeability, good stability, low cost, and low toxicity [15,16]. Various antimicrobial small molecules from natural products and synthetic compounds have been identified and developed. In this review, we aim to discuss antimicrobial small molecules against *S. mutans* based on the way they are developed, with a particular focus on their modes of action and mechanisms against the growth and virulence of *S. mutans*.

## 2. Drug Repurposing

Drug repurposing, also known as drug repositioning, is a commonly used drug development approach. Compared to new drug development, drug repurposing has many advantages including lower drug development cost, lower toxicity, and faster benchtop-to-clinic transition [17]. Besides, due to the long-term use of broad-spectrum antimicrobial agents, drug resistance is becoming increasingly prevalent in *S. mutans* [18]. Repositioning existing drugs as antibiotics is necessary for saving manpower and material sources. Small-molecule compounds exhibiting antimicrobial activity against other microorganisms have been widely screened for new uses against *S. mutans*. 

Screening FDA-approved drugs is an effective way to identify old drugs with new therapeutic effects against *S. mutans*. Saputo et al. screened 853 FDA-approved drugs and identified 126 candidates that exhibit antimicrobial activity against planktonic growth of *S. mutans*, among which 24 drugs inhibit biofilm formation, 6 drugs kill pre-existing biofilms, and 84 drugs exhibit both bacteriostatic and bactericidal effects against *S. mutans* biofilms. The 126 candidates were further classified into 6 categories, including antibacterials, ion channel effectors, antineoplastic drugs, antifungals, stains and disulfiram, many of which are small molecules such as biapenem, cefdinir, and zinc pyrithione [19]. Among the 126 candidates, a class of derivatives of the fat-soluble secosteroid vitamin D shows activity against *S. mutans*. One of the vitamin D derivatives, namely calcitriol, inhibits both planktonic cells and preforms *S. mutans* biofilms. Doxercalcierol, a synthetic vitamin D_2_ analog, reduces pre-existing biofilms and shows synergistic effects with bacitracin, a polypeptide that interferes with cell wall synthesis [20]. Gliptins is a common anti-human-dipeptidyl peptidase (DPP IV) drug for the treatment of type II diabetes. X-prolyl dipeptidyl peptidase (Sm-XPDAP) coded by the *pepX* gene is an analogous enzyme of DPP IV [21]. Sm-XPDAP plays a nutritional role in *S. mutans* [22]. The *pepX*-deficient strain of *S. mutans* produces fewer biofilms, suggesting that Sm-XPDAP is a potential target for the inhibition of *S.mutans* biofilms [23]. Considering the similarity between Sm-XPDAP and DDP IV, saxagliptin has been repurposed to inhibit *S. mutans*, which shows potent inhibitory effects on the biofilm formation of *S. mutans* [23]. 

Reserpine, another FDA-approved blood pressure medicine, has also been repurposed as an efflux pump inhibitor which suppresses acid tolerance and inhibits the glycosyltransferase activity of *S. mutans* and thus represents a promising treatment against cariogenic biofilms [24]. Screening drugs that target key metabolic processes is also commonly used. Folate metabolism is important for the syntheses of DNA, RNA, and amino acids in all organisms. Bedaquiline, an active drug firstly used to inhibit the ATP-synthase of mycobacteria [25], also shows a great antimicrobial activity against cariogenic bacteria in the acidic environment. In addition, bedaquiline can effectively inhibit the biofilm proliferation of oral pathogens, especially *S. mutans* [26].

Toremifene, an FDA-approved drug for the treatment of breast cancer, and zafirlukast, an antiasthma drug that has been approved in Europe and the USA, have also been repurposed to inhibit the growth and biofilm formation of *S. mutans* [27,28]. Another anticancer drug napabucasin (NAP), which is in phase III clinical trials for cancer treatment, shows antibacterial activity against *Escherichia coli*, *Streptococcus faecalis*, and *Staphylococcus aureus* [29,30]. Our group repurposed NAP against oral streptococci and found that NAP exhibits good antimicrobial activity against *S. mutans* biofilms [31]. In addition, by using NAP as a lead compound, we designed a novel small molecule, namely LCG-N25, which exhibits a good antibacterial activity and low cytotoxicity and induces no drug resistance of cariogenic *S. mutans* [32]. Repurposing existing antimicrobial drugs or antimicrobial groups is also a promising approach to the control of *S. mutans*. Nitrofuran has been reported to inhibit oral bacteria such as *S. mutans* and *Enterococcus faecalis* [33,34]. Based on the antimicrobial activity of nitrofuran against *S. mutans*, our group synthesized a novel water-soluble hybrid of indolin-2-one and nitrofuran, ZY354, which shows a good antimicrobial activity and selectivity against *S. mutans* [35]. Small molecules identified by drug repurposing are summarized in Table 1.

## 3. Screening from the Small-Molecule Library

Phenotypic screening is also a reliable approach to the identification of new antimicrobials. High-throughput screening from the small-molecule library is one of the main sources of phenotypic screening [36]. High-throughput drug screening based on probable target provides numerous compounds for further validation. PubChem, ZINC, DrugBank, ChemSpider, and MCE are the most popular databases, which contain bioinformatics data, cheminformatics data, and detailed targets of drugs [37]. The in silico screening of the compound library is an automatic method which can easily locate and optimize a lead compound. Molecular docking and molecular dynamic simulation are commonly used in in silico screening. Besides in silico screening, small molecules can be screened by culture-based approaches. 

*S. mutans* colonizes on the tooth surface and forms biofilms, which not only elevates its virulence, but also protects it from external influence such as antimicrobial treatment [38]. Key factors such as antigens I/II, glucosyltransferases (Gtfs), sortase A (SrtA), and quorum sensing (QS) systems are essential for *S. mutans* biofilms formation [39,40,41]. Screening small molecules against these biofilm-related factors is a promising strategy to identify new drugs that inhibit *S. mutans*. *S. mutans* adheres to the oral surface via two mechanisms, i.e., sucrose-independent and sucrose-dependent [42]. The sucrose-independent adhesion is mainly mediated by antigens I/II, which is also known as PAc [43,44,45], while the sucrose-dependent adhesion is mainly mediated by Gtfs including GtfB, GtfC, and GtfD [46], which also mediate the interspecies coaggregation and play a critical role in the development and maturation of oral biofilms [47,48]. Rivera-Quiroga et al. screened 883,551 molecules from the library “Small” and identified three molecules, namely ZINC19835187 (ZI-187), ZINC19924939 (ZI-939), and ZINC 19924906 (ZI-906), which inhibit *S. mutans* adhesion on polystyrene microplates by targeting antigens I/II [49]. Chen et al. screened a library of oxazole derivatives and identified a molecule called 5H6[2-(4-chlorophenyl)-4-{[(6-methyl-2-pyridinyl)amino]methylene}-1],3-oxazole-5(4H)-1, which is able to reduce the production of EPSs and inhibit *S. mutans* biofilms by inhibiting GtfC and GtfB [50]. Wu et al. screened a small-molecule library of 506 compounds and identified an active molecule, namely 2A4, which selectively inhibits *S. mutans* in multispecies biofilms modestly and inhibits both *S. mutans* planktonic cells and single-specie biofilms by downregulating virulence genes and inhibiting the production of antigens I/II and Gtfs [51]. The same group by using a structure-based virtual screening of 500,000 compounds against the GtfC catalytic domain identified a lead compound G43, which selectively bonds GtfC and thus inhibits the biofilm formation and cariogenicity of *S. mutans* [52]. Ren et al. also screened 15,000 molecules based on the structure of the *S. mutans* GtfC protein domain and found a quinoxaline derivative,2-(4-methoxyphenyl)-N-(3-{[2-(4-methoxyphenyl)ethyl]imino}-1,4-dihydro-2-quinoxalinylidene)ethanamine, which selectively bonds GtfC, reduces the synthesize of insoluble glucans and biofilms of *S. mutans* and thus inhibits the development of caries in vivo [53]. SrtA is a membrane-bound transpeptidase that anchors antigens I/II to the cell wall and thus contributes to the biofilm formation of *S. mutans* [41,54]. Samanli et al. screened 178 small molecules from a library and identified a SrtA inhibitor, namely CHEMBL243796 (kurarinone), which shows better a binding affinity with SrtA than CHX and exhibits a better pharmacokinetic activity toward *S. mutans* [55]. Luo et al. screened the ZINC library and the TONGTIAN library and identified several potential inhibitors of SrtA including benzofuran, thiadiazole, and pyrrole, which are able to bind to and inhibit SrtA. These SrtA inhibitors are promising for the control of *S. mutans* biofilms [56]. The QS system is a communication system that regulates *S. mutans* biological behaviors such as biofilm formation and dispersal [57,58]. Ishii et al. screened 164,514 small molecules against the peptidase domain of ComA, a key component of *S. mutans* QS, and identified 6 compounds that inhibit biofilm formation without repressing the cell proliferation of *S. mutans* [59]. 

Acid tolerance is another important phenotypic trait associated with the cariogenicity of *S. mutans* [60]. The proton pump F1F0-ATPase (H^+^-ATPase) is an important enzyme in the acid tolerance of *S. mutans* [61]. Sekiya et al. screened F1F0-ATPase inhibitors against *S. mutans* and found that piceatannol, curcumin, and desmethoxycurcumin (DMC; a curcumin analog) show marked activity against F1F0-ATPase of *S. mutans* and thus inhibit its growth and survival in acidic conditions, suggesting a potential anticaries strategy by inhibiting F1F0-ATPase [62].

In addition to the aforementioned molecules that have been proven to inhibit specific factors associated with the cariogenicity of *S. mutans*, an increasing number of small molecules have also been screened and identified to inhibit both planktonic cells and biofilms of *S. mutans*. Chen et al. screened about 2600 compounds from the MCE library and identified an antagonist of a calcium-sensing receptor, namely NPS-2143, which exhibits antimicrobial activity against methicillin-resistant *S. aureus* (MRSA) [63]. Further modifications of NPS-2143 yields a compound, namely II-6s, which shows a potent antimicrobial activity against both methicillin-resistant and methicillin-sensitive *S. aureus* [63]. Our group screened the derivatives of NPS-2143 and identified a small-molecule II-6s that effectively inhibits the growth and EPS generation of *S.*
*mutans*. In addition, II-6s shows lower cytotoxicity relative to CHX, significantly inhibits the demineralization of tooth enamel induced by *S. mutans* and induces no drug resistance in *S. mutans* after 15 passages [64], representing a promising alternative to the control of oral biofilms. Kim et al. synthesized a series of pyrimidinone or pyrimidindione-fused 1,4-naphthoquinones with antibacterial effects via pharmacophore hybridization, and they identified some derivatives with notable bacteriostatic and bactericidal effects against *S. mutans* in both resistant strains and sensitive strains [65]. Simon et al. [66] screened a library of 75 synthetic cyclic dipeptides (CDPs), which are a kind of stable metabolites from microorganisms [67], and identified 5 CDPs that inhibit *S. mutans* adhesion and biofilm formation. Zhang et al. screened a library containing 100 trimetrexate (TMQ) analogs and identified 3 compounds with selectively inhibitory effects against *S. mutans* [68]. Garcia et al. screened an antibiofilm library of 2-Aminoimidazole (2-AI) derivatives and identified a small molecule 3F1, which specifically disturbs *S. mutans* biofilms without dispersing biofilms of nonmutans *Streptococci* and reduces dental caries in rats [16]. Small molecules screened from molecule libraries are summarized in Table 2.

## 4. Screening from Natural Products

Natural products are an ample resource of drugs because of their structural diversity and biological activity [69]. Natural products and their derivatives accounted for about 32% of small-molecule drugs which are approved being on the market from 1981 to 2019 [70]. Natural products provide a large library for the identification of antimicrobials with lower cytotoxicity. 

Tea (*Camellia sinensis*) has many health benefits with antimicrobial, anti-inflammatory, and cancer-preventive activity [71,72]. The tea polyphenols epigallocatechin gallate (EGCG) has shown antimicrobial activity against *S. mutans* for decades. EGCG can inhibit the virulence of *S. mutans* including acid production, aciduricity, and biofilm formation. EGCG can reduce acid production of *S. mutans* by inhibiting the expression and activity of lactate dehydrogenase, suppress aciduricity by inhibiting F_1_F_0_-ATPase, and reduce the biofilm formation by inhibiting Gtfs activity and downregulating *gtf* genes [73,74,75]. A recent study investigated the effect of EGCG on the phosphoenolpyruvate-dependent phosphotransferase system (PEP-PTS) of both *S. mutans* and non-mutans streptococci and found that EGCG exhibits excellent inhibitory effects against the acid production of oral streptococci [76]. Melok et al. screened and identified a lipid-soluble green tea polyphenols based on EGCG, namely epigallocatechin-3-gallate-stearate (EGCG-S) with better stability and an antibiofilm activity equivalent to chlorhexidine gluconate [77]. In addition, the EGCG treatment showed lower cytotoxicity and better anti-inflammatory effects on *S. mutans*-stimulated odontoblast-like cells compared with CHX [78], indicating a potential application of EGCG to the management of dental caries.

Propolis is a hard, resinous, nontoxic natural product from plants with a history of being used as a dietary supplement. Propolis has shown a good antimicrobial activity against *S. mutans* for decades [79,80]. Koo et al. identified two small-molecule compounds from propolis extracts, namely apigenin and trans-trans farnesol (tt-farnesol), which exhibit distinguished biological activities against dental caries [81,82]. Apigenin, a 4β,5,7-trihydroxyflavone, can effectively inhibit Gtfs, specifically GtfB and GtfC. tt-farnesol, which is the most effective antibacterial compound in propolis, can reduce cell viability by disrupting membrane integrity and destabilizing oral biofilms rather than affecting Gtfs activities [81,83]. Moreover, tt-farnesol can reduce the intracellular iodophilic polysaccharides (IPS) accumulation of *S. mutans* and thus reduces the severity of smooth surface caries in rats [81,84]. The mechanism of tt-farnesol is likely attributed to the lipophilic moiety interaction with the bacterial membrane [84]. The combinatory use of apigenin, tt-farnesol, and fluoride can effectively reduce the biofilms and acidogenicity of *S. mutans* [84]. Caffeic acid phenethyl ester (CAPE), which is extracted from propolis, shows a broad-spectrum antimicrobial activity against *Enterococcus faecalis*, *S. aureus*, *Bacillus subtilis*, *Pseudomonas aeruginosa*, and other species [85]. A recent study has shown that CAPE not only affects the thickness of *S. mutans* biofilms, but also inhibits its biofilm formation and maturation, particularly by reducing EPS production [86,87]. 

In addition to the well-characterized tea catechins and propolis, other small molecules obtained from natural resources have also been shown to inhibit *S. mutans* planktonic cells and biofilms. He et al. showed that trans-cinnamaldehyde (TC) inhibited the acid production and aciduricity of *S. mutans* and downregulated virulence genes of *S. mutans* including *gtfD* [88]. Besides, TC showed synergistic effects with CHX on the inhibition of *S. mutans* biofilms and virulence by regulating genes related to metabolism, QS, bacteriocin expression, stress tolerance, and biofilm formation [89]. Ursolic acid has shown inhibitory effects on the EPS synthesis and the biofilm formation of *S. mutans* [90,91]. Resveratrol can inhibit the acid production, acid tolerance, and EPS production of *S. mutans* [92]. Ficin, a sulfhydryl protease isolated from the latex of fig trees, can inhibit the total protein and the biofilm formation of *S. mutans* and reduce the virulence of *S. mutans* [93]. Baicalin, another plant-derived molecule, can reduce the sucrose-dependent biofilm formation of *S. mutans* likely by inhibiting Gtfs. Baicalin can also downregulate virulence genes and inhibit the acid production of *S. mutans* [94]. Piceatannol, a kind of stilbene, can target the GtfC domain, inhibit glucans production and thus reduce *S. mutans* biofilm formation. Piceatannol can also inhibit *S. mutans* colonization in a sucrose-dependent drosophila colonization model [95]. β-sitosterol from Kemangi (*Ocimum basilicum* L.) can inhibit SrtA and thus suppresses *S.*
*mutans* biofilm formation [96]. Astilbin, a flavonoid from *Rhizoma Smilacis Glabtar*, can inhibit the activity of SrtA and the biofilm formation of *S. mutans* without repressing its growth [97]. Abietic acid, a natural product derived from pine rosin, also exhibits inhibitory effects on the acid production and the biofilm formation of *S. mutans* [98]. Rhodiola rosea, a traditional Chinese medicine, can inhibit the biofilm formation likely via downregulating *gtf* genes and genes associated with the QS system of *S. mutans* [99]. α-mangostin (αMG) extracted from tropical plants shows antimicrobial effects against planktonic cells of *S. mutans* [100] and can disrupt *S. mutans* biofilms by inhibiting the enzyme activity of GtfB, GtfC, and F1F0-ATPase [101]. N-arachidonoylethanolamine (AEA), a kind of endocannabinoids (ECs) [102], in combination with poly-L-lysine can inhibit *S.mutans* biofilm formation [103]. Small molecules screened from natural products are summarized in Table 3.

## 5. Target-Based Designing

Small molecules developed by target-based designing approaches can specifically inhibit *S. mutans*, which is expected to reduce the cariogenicity of oral biofilms without significantly disturbing other commensal bacteria. Key virulence factors of *S. mutans*, such as SrtA, antigens I/II, and Gtfs, are usually exploited as the targets for specific drug design. Small molecules designed by target-based approaches are summarized in Table 4.

### 5.1. SrtA and Antigens I/II Inhibitor

SrtA can catalyze antigens I/II and thus initiates the subsequent sucrose-independent adhesion and biofilm formation of *S. mutans* [42,54,97]. Recently, a series of SrtA inhibitors have been identified from natural products and synthetic compounds [104,105]. Many flavonoids have shown inhibitory effects on SrtA in Gram-positive bacteria [105]. A recent study using molecular docking demonstrated that myricetin is able to target the binding site of SrtA and thus inhibits SrtA activity and reduces the adhesion and biofilm formation of *S. mutans* [106]. Charles et al. synthesized several peptides spanning residues 803–185 of antigens I/II and identified a synthetic peptide p1025 that is able to inhibit antigens I/II binding to salivary receptors by forming an adhesion epitopes in a dose-dependent way. The study showed that Q1025 and E1037 of p1025 may be the two vital residues in the adhesion of p1025 toward antigens I/II. The effect of p1025 against *S. mutans* was tested by using a *Streptococcal* model in vitro, and p1025 shows moderate stability and selectivity against *S. mutans* recolonization to the tooth surface [107]. Li et al. also showed that dentifrice containing p1025 is able to prevent *S. mutans* recolonization in vitro and in vivo [108,109].

### 5.2. Gtfs Inhibitor

In the sucrose-dependent adhesion process, Gtfs synthesize EPS and allow *S. mutans* to adhere to oral surfaces and coaggregate with other microbes to form biofilms [110]. Molecules specifically targeting Gtfs can inhibit *S. mutans* biofilm formation and are promising for caries control. Flavonols show antibiofilm activities and inhibitory effects against *S. mutans* Gtfs [97,111]. Bhavitavya et al. screened a group of synthetic precursors of flavonols which consist of 14 hydroxychalcones, and several of them exhibit selectively effects against *S. mutans* biofilms. Based on compound 9 which is identified from a biofilm assay, 9b, a Z isomer of compound 9, shows better inhibition on *S. mutans*. 9b as a lead compound also exhibits selectivity against *S. mutans* biofilms by inhibiting Gtfs in a dose-dependent way [112]. Wu et al. screened and identified a Gtf inhibitor, namely G43, which showed notable effects on *S. mutans* biofilm formation [52]. Recently, this group further developed 90 analogs of G43 based on the structure activity relationship (SAR) of G43 and identified several new biofilm inhibitors with enhanced potency and selectivity. Different modifications based on G43 resulted in derivatives such as III_A6_, III_A8_, III_F1_, III_F2_, and III_F8_, which show an equally antibiofilm activity with G43 by inhibiting Gtfs. One of the leads compounds, III_F1_, selected after the comprehensive evaluation of SAR studies and zymogram results, can also inhibit *S.mutans* as a Gtf inhibitor, exhibit low toxicity to bacteria and have less effects on bacterial colonization compared to G43. The in vivo study showed a marked reduction of dental caries in rats, representing a promising adjuvant to the control of dental caries [113].

**Table 4 pathogens-10-01540-t004:** Small molecules designed by target-based approaches.

Small Molecules	Chemical Formula	Mechanisms	References
Compound III_F1_	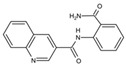	Selectively bond GtfC and significantly inhibit the biofilm formation	[113]
Myricetin (Myr)	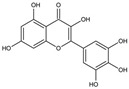	Inhibit SrtA and reduce the adhesion and biofilm formation of *S. mutans*	[106]
Peptide (p1025)		Inhibit the adhesion and biofilm formation of *S. mutans*	[107,108,109]
9b	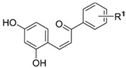	Inhibit *S.mutans* biofilms by inhibiting Gtfs	[112]

## 6. Conclusions

*S. mutans* is a well-recognized cariogenic species in the oral cavity. The effective inhibition or removal of this cariogenic bacterium is essential for the caries management. Small molecules are promising in this field due to their good antimicrobial activity, good selectivity, and low toxicity. Drug repurposing, drug screening from either small-molecule libraries or natural resources, and target-based designing are practical approaches to the development of small molecules that can effectively inhibit *S. mutans* and consequently benefit caries control. However, many issues have yet to be solved. First, the cytotoxicity of the novel molecules needs comprehensive evaluation before clinical translation, particularly for the synthetic molecules. Although drug repurposing has advantages such as lower cost, shorter development timelines, and relatively higher safety, how to reduce its known side effects and adverse reactions still needs further exploration. In addition, the application of reused drug is limited because of their original effects, and the indication of reused drugs is narrow compared to antibiotics. Second, although the mode of actions such as the inhibition of Gtfs and the suppression of acid production have been demonstrated for many small molecules, the underlying molecular mechanisms of these compounds are still not clear. Third, since oral biofilms consisted of numerous microorganisms, how to increase the selectivity of small molecules that specifically target *S. mutans* without interfering with other normal flora is one of the future directions for drug development. Specific inhibitors against *S. mutans* still need comprehensive validation in complex microbial consortia. Finally, the development of drug resistance by oral bacteria is still a concern that needs a long-term evaluation in both in vitro and in vivo models. Nevertheless, antimicrobial small molecules represent a promising approach to the effective inhibition of *S. mutans* and will benefit the management of dental caries. 

## Figures and Tables

**Table 1 pathogens-10-01540-t001:** Small molecules identified by drug repurposing.

Small Molecules	Chemical Structure	Mechanisms	References
Bedaquiline	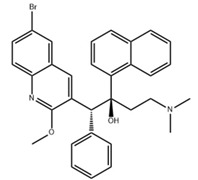	Inhibit cariogenic bacteria in an acidic environment and inhibit (*Streptococcus mutans*) *S. mutans* biofilm proliferation	[26]
Biapenem	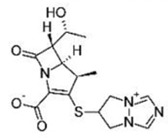	Inhibit the planktonic growth of *S. mutans;* inhibit *S. mutans* biofilms formation and reduce the viability of pre-existing *S. mutans* biofilms	[19]
Cefdinir	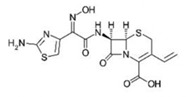	Inhibit the planktonic growth of *S. mutans;* inhibit the biofilm formation of *S. mutans*	[19]
Calcitriol	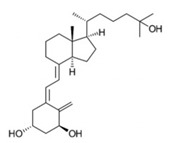	Inhibit the planktonic growth of *S. mutans;* reduce the viability of pre-existing *S. mutans* biofilms	[20]
Doxercalciferol	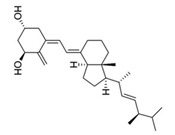	Inhibit the planktonic growth of *S. mutans;* inhibit the *S. mutans* biofilms formation and reduce the viability of pre-existing *S. mutans* biofilms	[20]
LCG-N25	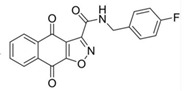	Inhibit both the planktonic cells and biofilms formation of *S. mutans*	[32]
Napabucasin	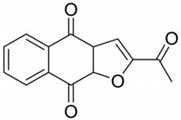	Inhibit *S. mutans* biofilms	[31]
Reserpine	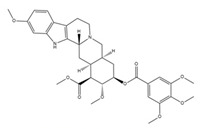	Suppress acid tolerance; inhibit the glycosyltransferase activity of *S. mutans*	[24]
Saxagliptins	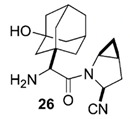	Reduce *S. mutans* biofilm formation	[23]
Toremifene	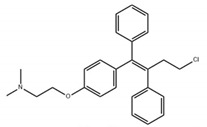	Inhibit the growth and biofilm formation of *S. mutans*	[27]
Zinc pyrithione	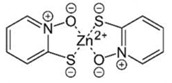		[19]
Zafirlukast	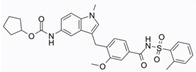	Inhibit *S. mutans* planktonic cells; inhibit biofilm formation and reduce the viability of the preformed biofilms of *S. mutans*	[28]
ZY354	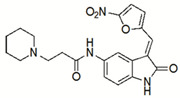	Inhibit *S. mutans* growth and selectively inhibit the biofilm formation of *S. mutans*	[35]

**Table 2 pathogens-10-01540-t002:** Small molecules screened from molecule libraries.

Small Molecules	Chemical Formula	Mechanisms	References
Compound 3F1	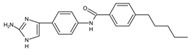	Specifically disturb *S. mutans* biofilms in a mixed biofilm	[16]
Compound 1	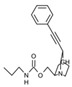	Inhibit biofilm formation by inhibiting quorum sensing systems	[59]
Curcumin	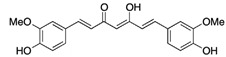	Inhibit F1F0-ATPase in *S. mutans* and inhibit *S. mutans* growth	[62]
Desmethoxycurcumin	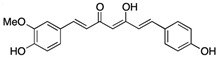
Piceatannol	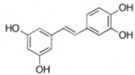
G43	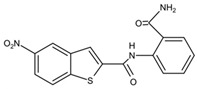	Inhibit *S. mutans* biofilm formation by selectively binding to GtfC	[52]
Pyrimidinone or pyrimidindione-fused 1,4-naphthoquinones	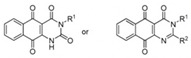	Bacteriostatic and bactericidal effects against *S. mutans* in both resistant and sensitive strains	[65]
ZINC19835187 (ZI-187)	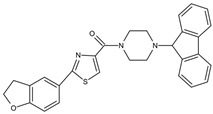	Inhibit *S. mutans* adhesion and biofilm formation by targeting antigens I/II	[49]
ZINC19924939 (ZI-939)	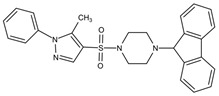
ZINC 19924906 (ZI-906)	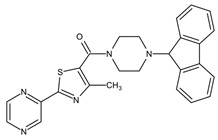
2A4	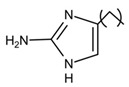	Inhibit *S. mutans* adhesion and biofilm formation by targeting antigens I/II and glucosyltransferases (Gtfs)	[51]
2-(4-methoxyphenyl)-N-(3-{[2-(4-methoxyphenyl)ethyl]imino}-1,4-dihydro-2-quinoxalinylidene) ethanamine	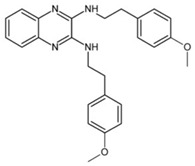	Inhibit the biofilm formation and destroy mature biofilms without killing *S. mutans* by inhibiting GtfC	[53]
II-6s	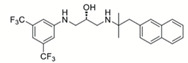	Inhibit growth and exopolysaccharides (EPS) generation of *S. mutans*;inhibit the demineralization of tooth enamel and induce no drug resistance in *S. mutans*	[64]
5H6	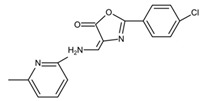	Inhibit the biofilm formation of *S. mutans* by antagonizing Gtfs	[50]

**Table 3 pathogens-10-01540-t003:** Small molecules screened from natural products.

Small Molecules	Chemical Formula	Mechanisms	References
Apigenin and trans-trans farnesol	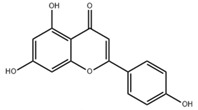	Inhibit Gtfs, specifically GtfB and GtfC; disrupt membrane integrity, destabilize oral biofilms and reduce the intracellular iodophilic polysaccharides (IPS) accumulation of *S. mutans*	[79,80,81,82,83,84,85,86,87]
Astilbin	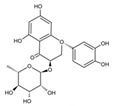	Inhibit the activity of sortase A (SrtA) and the biofilm formation of *S. mutans* without repressing its growth	[97]
Abietic acid	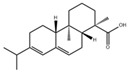	Inhibit acid production and the biofilm formation of *S. mutans*	[98]
AEA	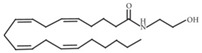	Show synergistic antibiofilm effects with poly-L-lysine aginst *S. mutans*	[103]
αMG	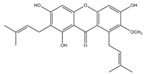	Disrupt *S. mutans* biofilms by inhibiting the enzyme activities of GtfB, GtfC, and F1F0-ATPase	[100,101]
Baicalin	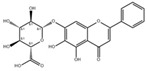	Inhibit biofilm formation, acid production, and virulence	[94]
β-sitosterol from Kemangi	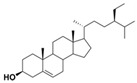	Inhibit *S. mutans* biofilm formation by inhibiting SrtA	[96]
Epigallocatechin gallate (EGCG)	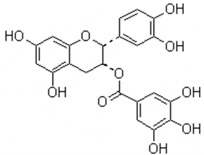	Inhibit *S. mutans* acid production, aciduricity, and biofilm formation	[73,74,75,76,77,78]
Piceatannol	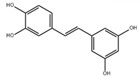	Inhibit glucans production by Gtfs, selectively affect *S. mutans* biofilms formation and inhibit *S. mutans* colonization in vivo	[95]
Resveratrol	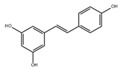	Inhibit acid production and aciduricity and down-regulated virulence genes	[92]
Trans-cinnamaldehyde (TC)	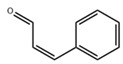	Inhibit virulence genes; show synergistic effects with CHX antibiofilms	[88,89]
Ursolic acid	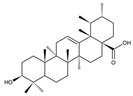	Inhibit biofilm formation and maturation by reducing EPS production	[90,91]

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
