# Peer review of "Small Molecule Compounds, A Novel Strategy against Streptococcus mutans"

_pathogens, 2021, doi:10.3390/pathogens10121540_

Round 1

Reviewer 1 Report

In this article, the author summarized several kinds of new small molecular compounds targeting S.mutans and biofilm, which showed us a promising approach for managing people with high risks of caries.

Minor concerns:

The English writing of this article should be more refined and accurate.

Page 2, line 59. “Comparing to …” should be “Compared to …” or “Comparing with”;

Page 11, reference 46 & page 14, reference 101. The format of references should be consistent with others.

Major concerns:

The writing logic of parts that discuss drugs in detail should be more integrated. For example, the sequences of new drugs can be declared by time of development or by the compounds sharing similar structures or modes of action.

Author Response

Thanks for your valuable comments.Please see the attachment.

Reviewer 2 Report

The manuscript describes several small molecules with anti-biofilm activity towards Streptococcus mutans, a major cariogenic bacterium. The manuscript is well written and the data are clearly presented. The are some issues that need to be addressed before the manuscript can be published according to the comments below.

Comments:

  • Line 63: Correct to Table 1.
  • Line 75: Provide some more detailed description of the role of Sm-XPDAP in mutans biofilms.
  • Line 106, Table 1: I would suggest to show the chemical formula of the various compounds. The same for compounds in Table 2 and 3. Also, provide the chemical structure of compounds mentioned in the text that are not mentioned in the Tables. Is there any structural- functional relationship for the various compounds with anti-biofilm activity towards mutans? Do these agents also act against other bacteria? Are these agents cytotoxic to mammalian cells? What are the effective doses?
  • Concerning the grammar in the Tables: For single agents, the verb should get the singular endings: -s/es.
  • How is SortaseA involved in biofilm formation? A reference should be provided to the sentence in line 136.
  • The action mechanisms of the different compounds mentioned ought to be described in more detail. Is there a common denominator for the action mechanism? E.g., Inhibition of GTFs/FTFs; inhibition of F-ATPase activity, effect on gene expression etc (Maybe add a Table for this).
  • Please mention which of the compounds can be used in the clinics. Maybe by adding an asterix in the Tables.
  • In Table 2 – some typo errors: correct to mutans (two places).
  • The last sentence of Table 2 should be in present form (similar to the other sentences).
  • Line 208: Correct to: F1F0-ATPase
  • Line 230: I think you mean biofilm formation (and not biofilm information).
  • Line 252: It is written "sucrose-dependent biofilm". Is there also another form for biofilm formation? Isn't sucrose used in the other studies?
  • Line 271: It is written "without significantly disturbing the commensal bacteria" What do you mean with commensal here? Do you intend viability?
  • Please explain in more depth " SrtA catalyzing surface protein Pac".
  • Line 304-305 – Please write the sentence in presence tense, to keep with the text style. Thus correct the sentence to: "G43, which selectively binds GtfC and significantly inhibits the biofilm formation and cariogenicity of mutans".
  • Line 320: Correct to "the synthetic molecules".
  • In conclusion: For drug repurposing, the drugs have already known pharmacological actions. Can these be compatible with the use as anti-cariogenic drugs? Please describe.
  • Some more anti-biofilm molecules can be added ti the manuscript such as thermophilins, curcumin, Bacillus subtilis metabolites, thiazolidiedione-8, endocannabinoids, cannabidiol, cannabigerol, curcumin, phosvitin, hainosan, Lactobacillus rhamnosus biosurfactant, sugar fatty acid esters etc.

Author Response

Thanks for your valuable comments. Please see the attachment.

Reviewer 3 Report

  1. The title needs to be changed. The focus of the paper is on repurposing drugs - "repurposing drugs" needs to be in the title. 
  2. The paper should have a clear rationale as to why repurposing drugs is warranted for Streptomyces mutans.
  3. The paper does not orient the reader; it downloads information but lacks context.

Author Response

(The authors gave the same response as above.)

Round 2

Reviewer 2 Report

Comments to revised version of pathogens-1429914

The manuscript has been improved, but there are still some issues that need to be addressed.

  1. Table 1: The authors write 125 candidate drugs – but at least some of them needs to be added to the Table.
  2. Table 1: The action mechanism of Zafirlukast needs to be added.
  3. Line 115: correct to adheres.
  4. Line 118: Correct to GtfD
  5. Line 125: Correct the parentheses [xx1].
  6. Line 139: I think you meant: "exhibited a better pharmacokinetic activity toward S. mutans." Please correct.
  7. Line 142: What do you mean with the sentence: "These new inhibtors showed a method in anti-S. mutans" Please provide a better explanation.
  8. Table 2 is incomplete – Please add the empty spaces with mechanisms and references. Also, what do you mean with "compound of PEP inhibitor"? Does it have a name?
  9. Line 9: Please correct to EGCG.
  10. Line 211: Instead of etc, please write "and other species".
  11. Line 213: Please correct to formation (instead of information).
  12. Line 219: etc is not a proper scientific word. Please specify.
  13. Table 3: Correct sentence to: "Showed synergistic anti-biofilm effect with CHX".
  14. Table 3: Remove comma after Astilbin.
  15. Table 3: Last row: Please correct the sentence to: "Show synergistic anti-biofilm effect with poly-L-lysine"
  16. In all tables – Maybe add the substances in alphabetic order and the names should be with an initial capital letter.
  17. Line 265: Correct to "synthesize".

Author Response

Dear reviewers,

Thanks for your valuable comments. According to your comments, we have carefully revised the full text, and the revised manuscript has been attached. In order to facilitate the review, the revised part of the text is marked up using the “Track Changes” function. The following is my reply to the reviewer's comments and the description of the revision of the original manuscript.

  1. Table 1: The authors write 125 candidate drugs – but at least some of them needs to be added to the Table.

Response: We have carefully checked and added the candidate drugs (e.g Calcitriol and Doxercalciferol) in the Table 1.

  1. Table 1: The action mechanism of Zafirlukast needs to be added.

Response: We have added the action mechanism of Zafirlukast in the table.

  1. Line 115: correct to adheres.

Response: We have corrected it as instructed.

  1. Line 118: Correct to GtfD

Response: We have corrected it as instructed.

  1. Line 125: Correct the parentheses [xx1].

Response: We have corrected it as instructed.

  1. Line 139: I think you meant: "exhibited a better pharmacokinetic activity toward S. mutans." Please correct.

Response: We have corrected it as instructed.

  1. Line 142: What do you mean with the sentence: "These new inhibitors showed a method in anti-S. mutans" Please provide a better explanation.

Response: We have rephrased this sentence for better clarity.

  1. Table 2 is incomplete – Please add the empty spaces with mechanisms and references. Also, what do you mean with "compound of PEP inhibitor"? Does it have a name?

Response: We have filled empty spaces and corrected compound name as “Compound 1” as it appeared in the referred literature accordingly.

  1. Line 9: Please correct to EGCG.

Response: We have corrected it as instructed.

  1. Line 211: Instead of etc, please write "and other species".

Response: We have corrected it as instructed.

  1. Line 213: Please correct to formation (instead of information).

Response: We have corrected it as instructed.

  1. Line 219: etc is not a proper scientific word. Please specify.

Response: We have added the specific content as instructed.

  1. Table 3: Correct sentence to: "Showed synergistic anti-biofilm effect with CHX".

Response: We have corrected it as instructed.

  1. Table 3: Remove comma after Astilbin.

Response: We have removed it as instructed.

  1. Table 3: Last row: Please correct the sentence to: "Show synergistic anti-biofilm effect with poly-L-lysine"

Response: We have corrected it as instructed.

  1. In all tables – Maybe add the substances in alphabetic order and the names should be with an initial capital letter.

Response: We have revised the table as instructed.

  1. Line 265: Correct to "synthesize".

Response: We have corrected his typo accordingly.